# Improving the Accuracy of Otitis Media with Effusion Diagnosis in Pediatric Patients Using Deep Learning

**DOI:** 10.3390/bioengineering10111337

**Published:** 2023-11-20

**Authors:** Jae-Hyuk Shim, Woongsang Sunwoo, Byung Yoon Choi, Kwang Gi Kim, Young Jae Kim

**Affiliations:** 1Department of Biomedical Engineering, Gil Medical Center, Gachon University College of Medicine, Incheon 21565, Republic of Korea; 2Department of Otorhinolaryngology-Head and Neck Surgery, Gil Medical Center, Gachon University College of Medicine, Incheon 21565, Republic of Korea; 3Department of Otorhinolaryngology-Head and Neck Surgery, Seoul National University Bundang Hospital, Seongnam 13620, Republic of Korea

**Keywords:** otitis media with effusion, otoendoscope, tympanic membrane, pediatric, artificial intelligence, deep learning, ResNet, DenseNet, Inception, InceptionResNet

## Abstract

Otitis media with effusion (OME), primarily seen in children aged 2 years and younger, is characterized by the presence of fluid in the middle ear, often resulting in hearing loss and aural fullness. While deep learning networks have been explored to aid OME diagnosis, prior work did not often specify if pediatric images were used for training, causing uncertainties about their clinical relevance, especially due to important distinctions between the tympanic membranes of small children and adults. We trained cross-validated ResNet50, DenseNet201, InceptionV3, and InceptionResNetV2 models on 1150 pediatric tympanic membrane images from otoendoscopes to classify OME. When assessed using a separate dataset of 100 pediatric tympanic membrane images, the models achieved mean accuracies of 92.9% (ResNet50), 97.2% (DenseNet201), 96.0% (InceptionV3), and 94.8% (InceptionResNetV2), compared to the seven otolaryngologists that achieved accuracies between 84.0% and 69.0%. The results showed that even the worst-performing model trained on fold 3 of InceptionResNetV2 with an accuracy of 88.0% exceeded the accuracy of the highest-performing otolaryngologist at 84.0%. Our findings suggest that these specifically trained deep learning models can potentially enhance the clinical diagnosis of OME using pediatric otoendoscopic tympanic membrane images.

## 1. Introduction

Otitis media with effusion (OME), also referred to as middle ear effusion (MEE), is a condition involving a build-up of fluid in the middle ear, usually stemming from an infection or inflammation disrupting the eustachian tube [1,2,3]. OME occurs most often in children following an upper respiratory infection or an ear infection, mainly due to shorter, undeveloped eustachian tubes being prone to blockages from bacterial infection or inflammation from recovery and irritants, such as allergens or smoke [4,5,6]. Symptoms of OME are usually painless, with reports of minor hearing loss and a feeling of fullness [3]. As such, there is difficulty in trying to diagnose OME with minor symptoms, and it can even go undiagnosed due to a lack of urgency in treating a painless condition [7]. While there is little risk of severe ear damage from untreated OME, particularly due to most children recovering from complications on their own, conductive hearing loss from persistent OME can risk affecting the child’s development of verbal communication and behavior even after OME resolves [8,9,10].

Traditionally, the diagnosis of OME involved the visual examination of the eardrum using otoscopes with illuminating magnifying lenses, providing a clear view of the ear canal for identifying fluid in the middle ear—a key diagnostic criterion of OME [11]. Additionally, otoscopes were often equipped with pneumatic bulbs that deliver a puff of air onto the tympanic membrane to assess for reduced mobility, a common characteristic of OME. However, the accurate diagnosis of OME using an otoscope can be challenging due to a lack of experience or training to accurately determine the features of OME, coupled with the uncooperative nature of the young children being diagnosed [12]. In an attempt to overcome these difficulties, modern otoendoscopes that can take clear digital images with a wide field of view were developed to provide access to real-time and recorded images of the tympanic membrane, even in young patients with narrow external auditory canals. Nonetheless, the diagnostic accuracies of otoendoscopes were still inconsistent, especially in inexperienced clinicians, due to differences in training and interpretation skills [13,14,15].

To alleviate the problems of subjective OME diagnosis, previous studies have explored the use of deep learning algorithms for classifying between healthy and OME using images of the tympanic membrane. A meta-review study showed that various studies were able to use a variety of neural networks, such as InceptionV3 and ResNet, to train on multiple otoendoscopic images for classification, reporting accuracies ranging from 76% to 97% [16,17,18,19,20,21]. However, deep learning OME diagnosis studies commonly utilized proprietary tympanic membrane images with varying acquisition protocols (otoscope, otoendoscope), preprocessing protocols (cropping, histogram equalization, augmentation), and evaluation metrics. Therefore, the application of external models to clinical settings can be challenging due to the potential use of images with different resolutions, angles, and formats in the training process than the images obtained with a clinic’s own imaging techniques and equipment, which can lead to unexpected results when classifying [16]. Additional validation studies using proprietary datasets obtained with personalized protocols may be beneficial for institutions considering their use in clinical practice. 

Most importantly, studies failed to differentiate between pediatric and adult patients, and they often lacked a direct comparison of deep learning model performances with those of experts and clinicians, making it difficult to evaluate the clinical potential of each trained model, especially when applied to pediatric patients [16]. Clarifying the use of pediatric data is particularly important due to major differences in the orientation of the tympanic membranes of small children and adults. The dimensions of the ear canal expand from 4.5 × 7.7 mm in children aged 5–8 to 5.4 × 8.6 mm in adults aged over 18, and the membrane slopes from the posterosuperior to the anteroinferior direction, allowing for a larger size compared to the ear canal [22,23]. As children mature, the growth of the skull base shifts the tympanic membrane toward a more vertical orientation [24]. Consequently, imaging acquired at a less perpendicular angle to the tympanic membrane’s plane using an otoendoscope in children could lead to apparent dimensional differences when compared with adults. As such, it is vital that the classification models are trained on pediatric tympanic membrane images, given the substantial anatomical differences from adult images and considering that the primary application of the model is the diagnosis of otitis media with effusion predominantly in pediatric patients.

For this study, we prepared 1150 pediatric tympanic membrane otoendoscopic images differentiated by control and OME, then we utilized various deep learning models, such as ResNet50, DenseNet201, InceptionV3, and InceptionResNetV2, to train a neural network to distinguish critical features of OME in asymptomatic pediatric patients for the differential diagnosis of OME. A dataset of 100 additional images separate from the 1150 images was set aside to compare the performance of the trained deep learning model with the diagnostic performance of otolaryngologists. We aimed to validate the diagnostic performance of a deep learning model trained on proprietary tympanic membrane images then evaluate the model’s potential for aiding the clinical diagnosis of OME.

## 2. Materials and Methods

### 2.1. Dataset

The institutional review boards of the Clinical Research Institute at our medical center approved this study (IRB No. B-1905/540-114) and waived the requirements for informed consent, considering the retrospective study design and the use of anonymized patient data. Data were collected from an electronic medical records database and analyzed anonymously. All methods employed in this study were in accordance with the approved guidelines and the Declaration of Helsinki.

A total of 1250 tympanic membrane images were collected from otoendoscopic examinations of pediatric patients under 15 years of age, following the Korea-based clinical practice guideline recommendations for pediatric OME [25]. These images were obtained using a 0° straight telescope (diameter of 3 mm, Karl Storz, Tullingen, Germany) during examinations for suspected OME or other clinical conditions with a normal tympanic membrane, such as sudden sensorineural hearing loss. The inclusion of such conditions was to enrich our dataset for the model’s training and make it more representative of the diverse clinical cases that might be encountered. However, this condition was carefully selected, given that it is less common in this age group compared to adults.

Each image was thoroughly examined before inclusion. Low-quality blurry images from motion artifacts, out-of-focus images, and images with more than half of the tympanic membrane obscured by cerumen were excluded to ensure that sufficient quality images were used for training. For the images of normal tympanic membranes, diagnoses were confirmed by experienced otolaryngologists to ensure the accuracy of the “normal” label. For the OME group, only images from ears with confirmed middle ear effusion by myringotomy were included, serving as a reference standard for the diagnosis of OME. This rigorous selection process aimed to ensure the reliability and validity of our training data, which is essential for the effective training of our deep learning model. 

The training/validation/testing datasets contained 592 OME and 558 normal tympanic membrane images. Of these 1150 images, 920 images were used for the training/validation datasets, consisting of 474 OME images and 446 normal images, which were split into a ratio of 8:2 for training (380 OME, 357 normal) and validation (94 OME, 89 normal). A total of 230 images were used for the test dataset, which consisted of 118 OME images and 112 normal images. The 920 training images and 230 test images were randomly selected for each cross-validation fold. An additional 50 OME and 50 normal images separate from the training/validation dataset were set aside to compare deep learning model diagnostic performances with those of 7 otolaryngologists. An example of data allocation is shown in Figure 1. 

### 2.2. Development Environment

The system for deep learning consisted of 4 NVIDIA TITAN Xp (NVIDIA, Santa Clara, CA, USA) graphics processing units (GPUs), a Xeon E5-1650 v4 (Intel, Santa Clara, CA, USA) central processing unit (CPU), and 128 GB of random access memory (RAM). Deep learning was conducted using Python 2.7.6 and the Keras 2.1.5 framework with a TensorFlow backend in the Ubuntu 14.04 operating system.

### 2.3. Preprocessing

Unprocessed tympanic images taken by otoendoscopes were comprised of the tympanic membrane cropped in a circular shape with a black background. Due to the inconsistent nature in which the images were captured, tympanic images were often uncentered and varied in size. To preprocess each tympanic image, the minimum and maximum non-zero x and y values of each circular tympanic image were obtained to crop out the unnecessary background, maximizing the view of the tympanic membrane while normalizing each image into a square shape. The cropped images were resized to a resolution of 256 × 256 then histogram-equalized using contrast-limited adaptive histogram equalization (CLAHE). In order to improve the number of training data used for this study, the preprocessed training images were augmented for 20 batches, with a rotation range of 5, width and height shift range of 0.05, shear range of 0.05, and zoom range of 0.05. A total of 17,468 images were generated for training data.

### 2.4. Deep Learning

The deep learning classification neural network architectures used to train on tympanic membrane images for diagnosis were ResNet50 [26], DenseNet201 [27], InceptionV3 [28], and InceptionResNetV2 [29]. The traditional convolutional neural network that the models are based on learns to map input data (like images) to the desired output (like class labels) through a series of convolutional layers [30]. Each layer learns a part of this mapping by adjusting its weights during training through a process known as backpropagation. However, as the network gets deeper, the gradient (used to update the weights) can become very small, leading to the vanishing gradient problem where gradients shrink as they backpropagate through deep networks and make it hard to train layers at the beginning of the network [31]. This makes it difficult for layers, especially the earlier ones, to learn effectively.

The Residual Network (ResNet) is a deep neural network that primarily focuses on solving the vanishing gradient problem [26]. By introducing skip connections, also referred to as residual connections, ResNet allows gradients to flow directly back through these connections, enabling the training of deeper networks while mitigating the degradation problem. The residual connections also allow the network to establish an identity function that ensures the higher layer performs as well as the lower layer without degrading in performance. This alleviates the pressure of the model having to learn an end-to-end mapping directly and allows the stacked layers to learn more refined and complex mappings. Additionally, by combining direct signals with the outputs from the intermediate layers, the network is better able to preserve the gradient magnitude throughout the training process, facilitating learning across all layers.

The Dense Network (DenseNet) is a deep neural network architecture designed to combat the vanishing gradient problem through a dense connectivity pattern among layers that receives additional inputs from all preceding layers and passes on its own feature-maps to all subsequent layers [27]. Each layer in DenseNet is connected to every other layer in a feed-forward fashion, meaning that the feature maps learned by any layer are directly accessible to all subsequent layers. This creates an environment where information and gradients can be communicated much more effectively through the network. The densely connected design of DenseNet also addresses the issue of vanishing gradients by ensuring that each layer has direct access to the gradients from the loss function and the original input signal, leading to implicit deep supervision during learning. The connectivity also allows feature maps to be reused throughout the network, significantly reducing the number of parameters, as there is no need to relearn redundant feature-maps.

InceptionV3 is a deep neural network architecture with signature inception modules to perform convolutions at multiple scales concurrently, allowing the model to handle different aspects of the image with filters that are appropriate for each scale [28]. InceptionV3, the third iteration of the inception architecture, refines this approach by factorizing convolutions into smaller, more manageable operations, making the network faster and also reducing the number of parameters. InceptionV3 also incorporates auxiliary classifiers to propagate label information lower down in the network for additional regularization and applies a grid size reduction technique to reduce the dimensions of the grid representation, effectively increasing the depth and width of the network without a significant increase in the computational cost.

InceptionResNetV2 is a neural network model that utilizes the multi-scaling inception architecture in tandem with the residual connections [29]. The inception modules allow the network to choose which scale to emphasize in each part of the image, while the residual connections counteract the vanishing gradient by providing a direct path for the gradient during backpropagation. The hybrid model using residual connections allows for a deep and wide architecture with a significantly accelerated training process while also providing a rich feature extraction capability through the inception modules. The combination of the two architectures enhances the learning of high-level and low-level features for accurate classification.

To achieve improved performances from the deep learning models, pretrained ImageNet weights trained on approximately 1.3 million images from the ImageNet dataset were used for transfer learning, augmenting parameters acquired from the dataset of the training model [32]. Additionally, 5 k-fold cross-validation was performed for each model with randomized training and testing datasets (maintaining the same abnormal, normal ratio within each training and testing dataset) to detect overfitting. Each k-fold cross-validated model was trained with the input size of 256 × 256 × 3, for 250 epochs, with a batch size of 40, using the Adam optimizer, a categorical cross-entropy loss function with a learning rate of 0.0001.

### 2.5. Observer Study by Otolaryngologists

To measure interobserver agreement between the deep learning algorithm and otolaryngologists, 100 tympanic membrane images with varying degrees of diagnostic difficulty were randomly selected from the 1250 image dataset. This test set was separated from the training and validation sets. Then, 2 invited otologists with 11 to 19 years of experience and 5 residents with 4 to 5 years of clinical otolaryngology training (3 fifth-year residents and 2 fourth-year residents) were asked to label each otoendoscopic tympanic membrane image to determine whether middle ear effusion was present or absent. Since clinical information was not provided to the invited observers, they were not asked to distinguish acute otitis media from OME. While reviewing the images, the invited radiologists were also requested to rate the diagnostic confidence level of their OME diagnosis based on a 6-point ordinal scale: 1, definitely not OME; 2, probably not OME; 3, possibly not OME; 4, possibly OME; 5, probably OME; and 6, definitely OME. The accuracy, sensitivity, and specificity of the results were determined from such data.

### 2.6. Statistical Analysis

The accuracy, sensitivity, specificity, and area under the receiver operating characteristic curve (AUC) of each k-fold model trained through the deep learning algorithm were measured. In the comparisons between deep learning models and observers, the accuracy and AUC of each k-fold model were calculated using the probability predicted by the deep learning, whereas the accuracies and AUCs of radiologists were determined using their diagnostic confidence level of OME.

### 2.7. Gradient-Weighted Class Activation Mapping

Gradient-weighted Class Activation Mapping (Grad-CAM) was utilized to visually explain the decisions made using CNN-based models [33]. Grad-CAM produced heatmaps overlaid on top of tympanic membrane images that designated the regions on each image that were important for prediction.

## 3. Results

### 3.1. Model Performance

We evaluated the performances of each cross-validated k-fold ResNet50, DenseNet201, InceptionV3, and InceptionResNetV2 model with accuracy, sensitivity, and specificity as shown in Table 1. Across 5 k-folds, ResNet50 showed an average accuracy of 92.9%, specificity of 86.5%, sensitivity of 99.2%, and AUC of 0.992. DenseNet201 showed an average accuracy of 97.2%, specificity of 95.2%, sensitivity of 99.6%, and AUC of 0.999. InceptionV3 showed an average accuracy of 96.0%, specificity of 95.4%, sensitivity of 97.0%, and AUC of 0.995. InceptionResNetV2 showed an average accuracy of 94.8%, specificity of 92.2%, sensitivity of 98.8%, and AUC of 0.998.

### 3.2. Comparison between Model Diagnostic Performance and Otolaryngologist Diagnostic Performance

Each k-fold model performed predictions on 100 images (50 OME, 50 normal) unexposed to the deep learning model for comparison with the diagnostic performance of otolaryngologists. The side-to-side accuracy comparisons between each trained model and otolaryngologist are shown in Table 2. The mean accuracies and AUCs of the trained models were 92.9% and 0.992 (ResNet50), 97.2% and 0.999 (DenseNet201), 96.0% and 0.995 (InceptionV3), and 94.8% and 0.998 (InceptionResNetV2), while the accuracies and AUCs of physician observers were 81.0% and 0.845, 84.0% and 0.863, 74.0% and 0.743, 75.0% and 0.773, 69.0% and 0.798, 70.0% and 0.727, and 70.0% and 0.737. The order of observers is ordered from the most experienced (1–2: faculty members in otology) to the least (3–5: fifth-year residents; 6–7: fourth-year residents). The highest accuracy performance of physician observers was 84.0%, which did not exceed the average accuracy of the lowest-performing model (ResNet50, 92.9%) nor the accuracy of the worst-case scenarios of ResNet50 trained on k-fold 3 (88.7%), DensNet201 trained on k-fold 5 (95.0%), InceptionV3 trained on k-fold 4 (89.0%), and InceptionResNetV2 trained on k-fold 3 (88.0%).

### 3.3. Grad-CAM Heatmaps

Heatmaps illustrating the focal points of tympanic membrane images utilized in prediction overlaid on top of tympanic images are shown in Figure 2. The intense portions of the heatmap (colored in red) seemed to focus on the anterior inferior quadrant of the tympanic membrane (the most gravitationally dependent area of the middle ear where fluid accumulates) in determining both normal and OME images.

## 4. Discussion

For this study, we utilized 1150 pediatric tympanic membrane images to train and test multiple cross-validated deep learning neural networks for diagnosing OME. The correct diagnosis of OME can be heavily dependent on the experience of the observer, based on factors such as the training received, specializations, or years of practice [15,34]. In order to address the issues of inconsistent OME diagnostic accuracies in inexperienced physicians, previous studies have utilized various forms of deep learning models trained on tympanic membrane images to develop tools to assist OME diagnosis [16]. However, few studies have specified the use of pediatric images in the training of these models, making it difficult to verify their effectiveness in clinical OME diagnosis where pediatric patients are prevalent [16]. Due to multiple differences in tympanic membrane images acquired from young children and adults, it is important that training classification models for OME, a disease that occurs mostly in children, be trained using pediatric images [22,23,24]. As such, we aimed to train deep learning classification models for OME diagnosis using pediatric tympanic membrane images to aid the clinical diagnosis of OME in pediatric patients. In our study, the cross-validated deep learning models trained on pediatric tympanic membrane images attained mean accuracies and AUCs of 92.9% and 0.992 (ResNet50), 97.2% and 0.999 (DenseNet201), 96.0% and 0.995 (InceptionV3), and 94.8% and 0.998 (InceptionResNetV2). Additionally, we compared the diagnosis performances of the models trained on pediatric images with the diagnostic performance of multiple otolaryngologists. Every model showed higher average diagnostic accuracies (92.9%, 97.2%, 96.0%, 94.8%) and AUCs (0.992, 0.999, 0.995, 0.998) than the highest diagnostic accuracy (84%) and AUC (0.863) of otolaryngologists. Results showed that our diagnostic model can outperform otologists who specialize in ear disorders and assist in diagnosing OME in pediatric patients using tympanic membrane images obtained with an otoendoscope.

Various studies have utilized a variety of deep learning models to classify OME using tympanic membrane images with robust accuracies. A study that trained on 267 pediatric intraoperative ear images and used ResNet34 with ImageNet transfer learning managed to reach an OME diagnostic accuracy of 83.8% despite its low training sample size of 270 images [21]. Other studies that utilized similar ResNet variants (ResNet101, ResNet50, ResNet18) that trained on proprietary otoendoscopic images achieved accuracies of 91.7%, 93.4%, and 97.2% but did not specify whether the images used to train and test the models were pediatric [17,19,20]. Other deep learning architectures aside from ResNet were also utilized to train models that can classify OME using tympanic membrane images. InceptionV3 deep learning models were also used to train and classify OME with accuracies of 92.1% and 76.0% [17,18]. MobileNetV2, a mobile architecture designed to operate on smart devices, was able to classify using pediatric otoscopic images transmitted to smartphones through Wi-Fi with an accuracy of 83.3% [20]. Our average accuracies of ResNet50, DenseNet201, InceptionV3, and InceptionResNetV2 models (92.9%, 97.2%, 96.0%, 94.8%) trained and validated using pediatric images were well within or exceeded the ranges of previously reported studies.

In order to assess the possibility of utilizing the trained models for clinical usage, the diagnostic accuracies of each model were compared with the diagnostic performances of otolaryngologists. Otolaryngologists on average falsely diagnosed OME in 16.9 images and normal in 8.4 images out of each 50 tested images, while the deep-trained model on average falsely classified OME in 0.8 images and normal in 5.4 images. False diagnoses of OME were more common in observers than in our trained models, likely due to the selective bias of the observers being averse to incorrectly selecting false negatives. Confusion in diagnosis can arise from the attentive focus on salient lesion areas that are identified as OME biomarkers, which can be commonly misidentified when observing key characteristics of the tympanic membrane [20]. The results of our Grad-CAM analysis, represented in Figure 2, indicate that the deep learning networks placed significant weight on the anterior inferior quadrant of the tympanic membrane when determining the correct diagnosis of OME. A focus on such areas seemed to contribute to a higher classification accuracy compared to human experts, potentially due to the networks’ ability to more effectively distinguish between patterns that are commonly misidentified as normal or OME by otolaryngologists. When comparing the performance of our models in diagnosing OME with the performance of otolaryngologists, the models demonstrated higher average accuracies (92.9%, 97.2%, 96.0%, 94.8%) in diagnosing OME. Even in the worst-case scenarios of each model (ResNet50: 88.7%; DenseNet201: 95.0%; InceptionV3: 89.0%; InceptionResNetV2: 88.0%), all of them performed higher than the best-performing observer with an accuracy of 84.0%. The results of our cross-validated deep learning models suggest that these models have the potential to improve the accuracy of OME diagnosis in clinical settings, particularly in situations where the diagnosis may be made by untrained pediatricians with lower diagnostic accuracy than otolaryngologists.

There are several limitations to this study. First, while our study utilized a sufficient number of images, they were acquired using the same type of otoendoscope, reducing the generalization of the dataset. As such, it is possible that the trained model shows different diagnostic accuracy in a clinical setting where a different otoendoscope and image format can be used to acquire images for diagnosis. The potential difference in diagnostic accuracy can be further exacerbated if the otoendoscopes utilized by untrained clinicians are equipped with poor resolution or lighting. Future deep learning studies with tympanic membrane images acquired with various types of endoscopes and imaging formats could help alleviate this problem. Second, the data that were selected for this model only consisted of high-quality images of tympanic membranes that were cerumen-free. As such, the diagnostic performance of the model in cases where otoendoscopies are performed by untrained clinicians on tympanic membranes that are obstructed and cerumen-filled may be lower than the results shown in this study. Third, obtaining high-quality images of the tympanic membrane in young patients proved challenging due to their uncooperative nature. To overcome this obstacle, multiple images of the same tympanic membrane were captured, and the image of superior quality was selected for inclusion in the dataset used in this study. To address this issue in future research, endoscopic recordings of the tympanic membrane could be employed to train and evaluate deep learning models. Fourth, the tympanic membrane images were obtained predominantly from Korean children, and potential differences in the tympanic membranes of different populations were not accounted for. Future studies with a wider distribution of tympanic membrane images obtained from various different populations world-wide could help improve the generality of the trained classification models.

## 5. Conclusions

Our study was able to utilize ResNet50, DenseNet201, InceptionV3, and InceptionResNetV2 to train deep learning networks for differentially diagnosing OME in pediatric tympanic membrane images. The trained networks showed better OME diagnostic accuracies on average than the diagnostic performance of seven otolaryngologists, showing their potential for assisting in the clinical diagnosis of OME in pediatric patients.

## Figures and Tables

**Figure 1 bioengineering-10-01337-f001:**
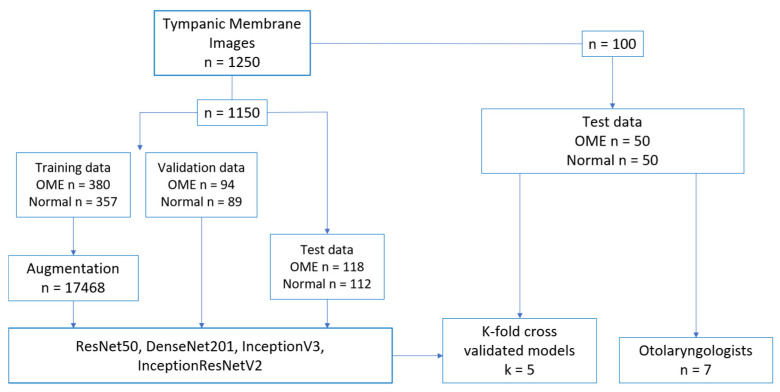
Data allocation of tympanic membrane images.

**Figure 2 bioengineering-10-01337-f002:**
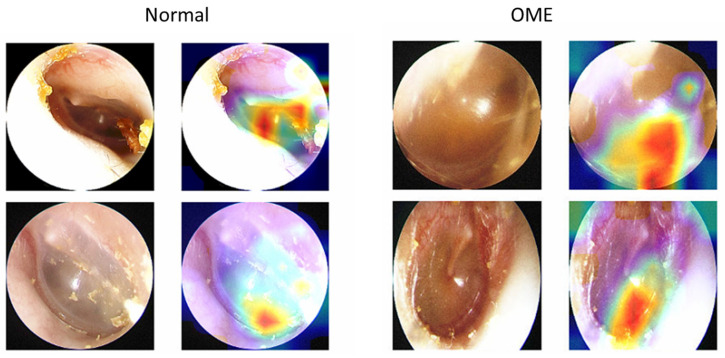
Class Activation Mapping (CAM) heatmap images of correctly predicted normal and OME tympanic membrane images. Heatmap overlaid on top of each tympanic membrane with a more intense color (red) indicates higher activation.

**Table 1 bioengineering-10-01337-t001:** Accuracy, sensitivity, specificity, and AUC of each k-fold cross-validated model.

Model	k-Fold ^1^					
ResNet50	1	2	3	4	5	Avg.
Accuracy	96.8%	95.6%	88.7%	90.0%	93.5%	92.9%
Specificity	95.6%	92.0%	77.7%	81.3%	85.7%	86.5%
Sensitivity	100.0%	99.2%	99.2%	97.5%	100.0%	99.2%
AUC	1.000	0.993	0.994	0.972	1.000	0.992
DenseNet201	1	2	3	4	5	
Accuracy	99.0%	100.0%	97.0%	95.0%	95.0%	97.2%
Specificity	98.0%	100.0%	96.1%	90.9%	90.9%	95.2%
Sensitivity	100.0%	100.0%	98.0%	100.0%	100.0%	99.6%
AUC	1.000	1.000	0.996	1.000	0.998	0.999
InceptionV3	1	2	3	4	5	
Accuracy	98.0%	99.0%	97.0%	89.0%	97.0%	96.0%
Specificity	97.0%	99.0%	95.0%	89.8%	96.1%	95.4%
Sensitivity	100.0%	99.0%	100.0%	89.0%	98.0%	97.0%
AUC	0.999	1.000	0.994	0.986	0.998	0.995
InceptionResNetV2	1	2	3	4	5	
Accuracy	100.0%	97.0%	88.0%	92.0%	97.0%	94.8%
Specificity	100.0%	94.3%	80.6%	86.2%	100.0%	92.2%
Sensitivity	100.0%	100.0%	100.0%	100.0%	94.0%	98.0%
AUC	1.000	0.995	0.999	1.000	0.997	0.998

^1^ Each k-fold column represents the performance of each k-fold trained model evaluated on the 100-image dataset unexposed to the trained models.

**Table 2 bioengineering-10-01337-t002:** Comparison of diagnostic performance between the k-fold averages of each model and each individual otolaryngologist.

Trained Model Averages							
	ResNet50		DenseNet201	InceptionV3		InceptionResNetV2	
Mean Accuracy	92.9% (±3.1%)		97.2% (±2.0%)	96.0% (±2.0%)		94.8% (±4.3%)	
Worst ^1^	88.7%		95.0%	89.0%		88.0%	
Best ^2^	96.8%		100.0%	99.0%		100.0%	
Mean AUC	0.992 (±0.010)		0.999 (±0.002)	0.995 (±0.005)		0.998 (±0.002)	
Observer Judgement							
Observer	1	2	3	4	5	6	7
Accuracy	81.0%	84.0%	74.0%	75.0%	69.0%	70.0%	70.0%
AUC	0.845	0.863	0.743	0.773	0.798	0.727	0.737

^1^ Worst refers to the lowest accuracy among the 5 k-folds (of each respective model type) displayed when evaluating the performance of each k-fold trained model on the 100-image dataset unexposed to the trained models. ^2^ Best refers to the highest accuracy among the 5 k-folds. Values in parentheses refer to the standard deviation of the mean.

## Data Availability

The code and dataset are available on reasonable requests from the corresponding author.

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
