# Peer review of "Improving the Accuracy of Otitis Media with Effusion Diagnosis in Pediatric Patients Using Deep Learning"

_bioengineering, 2023, doi:10.3390/bioengineering10111337_

Round 1
Reviewer 1 Report
Comments and Suggestions for Authors
The authors aim to explore deep learning networks to aid in diagnosing OME. Their findings suggest that specifically trained deep learning models can aid the clinical diagnosis of OME using pediatric otoendoscopic tympanic membrane images.
This is an interesting manuscript. Minor suggestions
1. Please explain how this study adds to the current knowledge in comparison to the other studies available.
2. Is the data comparable across the population worldwide?
3. The accuracy of varying otoendoscopes used. Justify this.
Author Response
- Please explain how this study adds to the current knowledge in comparison to the other studies available.
While there have been previous studies that explored the use of classification models to diagnose OME, our study aims to address the common problems that most studies showed, most importantly regarding the use of pediatric tympanic membrane images. Most studies did not identify if they used images of pediatric patients, and due to differences in tympanic membrane shape between that of a child and an adult, it is important that training a model for a disease that occurs mostly in children be trained on pediatric images. Additionally, we validated the performance of our trained models with otolaryngologists with varying levels of experience, something that was missing in many studies.
We added a paragraph in the introduction (paragraph 4, page 2) detailing important differences between the tympanic membrane image of a child and an adult.
- Is the data comparable across the population worldwide?
Unfortunately, we could not compare the data and results using tympanic membrane images from different populations due to a lack of access. We added this to the limitations section (last paragraph of discussion, page 9)
- The accuracy of varying otoendoscopes used. Justify this.
Only one type of otoendoscope was used for this study, but we’ve added a limitation that the use of only a single type of otoendoscope can affect the generalization of the trained model (last paragraph of discussion, page 9).
Reviewer 2 Report
Comments and Suggestions for Authors
In this study, we performed extensive training and cross-validation of ResNet50, DenseNet201, InceptionV3 and InceptionResNetV2 models using a dataset of 1150 pediatric tympanic membrane images obtained through otoendoscopy. These models were specifically trained to classify Otitis Media with Effusion (OME). Subsequent evaluation on an independent dataset of 100 pediatric tympanic membrane images revealed remarkable performance with average accuracies of 92.9% (ResNet50), 97.2% (DenseNet201), 96.0% (InceptionV3) and 94.8% (InceptionResNetV2). This contrasts with the diagnostic accuracy achieved by otolaryngologists, which ranges from 84.0% to 69.0%. The results highlight the potential of these trained deep learning models to improve the clinical diagnosis of OME through the analysis of pediatric otoendoscopic tympanic membrane images.
In this regard, this study effectively addresses an important gap in the existing literature in this field. However, it is recommended that the abstract be improved by removing the general introductory sentences from the abstract and placing more emphasis on the findings and motivation of the study. I would also suggest explaining the algorithm of the deep learning model and providing a detailed description of the mathematical model used in the deep learning section.
Author Response
- Removing the general introductory sentences from the abstract and placing more emphasis on the findings and motivation of the study.
A sentence was removed from the general introduction of the abstract and replaced with the motivation emphasizing the importance of using pediatric tympanic membrane images due to specific distinctions between a child and an adult tympanic membrane. An additional line comparing the accuracy of the trained model and the otolaryngologist was added to the abstract as well.
- I would also suggest explaining the algorithm of the deep learning model and providing a detailed description of the mathematical model used in the deep learning section.
Four new paragraphs describing each segmentation model used (ResNet, DenseNet, InceptionV3, InceptionResNetV2) was added to the deep learning section (page 4-5)